# An empirical study of software ecosystem related tweets by npm maintainers

Syful Islam[1,*], Yusuf Sulistyo Nugroho[2,*], Chy. Md. Shahrear[3], Nuhash Wahed[3], Dedi Gunawan[2], Endang Wahyu Pamungkas[2], Mohammed Humayun Kabir[3], Yogiek Indra Kurniawan[4] and Md. Kamal Uddin[3]

[1] Bangabandhu Sheikh Mujibur Rahman Science and Technology University, Gopalganj, Bangladesh
[2] Universitas Muhammadiyah Surakarta, Surakarta, Indonesia
[3] Noakhali Science and Technology University, Noakhali, Bangladesh
[4] Universitas Jenderal Soedirman, Purwokerto, Indonesia
* These authors contributed equally to this work.

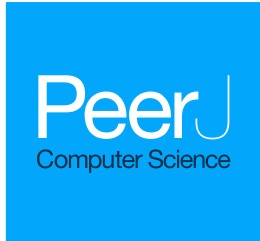

Corresponding authors
Syful Islam,
syfulcse@bsmrstu.edu.bd
Mohammed Humayun Kabir,
kabir@nstu.edu.bd

## ABSTRACT

The npm ecosystem is crucial for the JavaScript community and its development is significantly influenced by the opinions and feedback of npm maintainers. Many software ecosystem maintainers have utilized social media, such as Twitter, to share community-related information and their views. However, the communication between npm maintainers *via* Twitter in terms of topics, nature, and sentiment have not been analyzed. This study conducts an empirical analysis of tweets by npm maintainers related to the software ecosystem to understand their perceptions and opinions better. A dataset of tweets was collected and analyzed using qualitative analysis techniques to identify the topic of tweets, nature, and their sentiments. Our study demonstrates that most tweets belong to the package management category, followed by notifications and community-related information. The most frequently discussed topics among npm maintainers in the package management category are usage scenarios. It appears that the nature of tweets mostly shared by npm maintainers is information, followed by question and answer, respectively. Additionally, the sentiment analysis reveals that npm maintainers express more positive sentiments towards notification and community-related discussion while expressing more neutral opinions towards the package management related discussion. This case study provides valuable insights into the perceptions and opinions of the npm maintainers regarding the software ecosystem and can inform future development and decision making.

## INTRODUCTION

Third-party packages are now an integral part of contemporary software development, especially in building web or mobile applications (*Decan, Mens & Grosjean, 2019*; *Serebrenik & Mens, 2015*; *Abdalkareem et al., 2017*). In almost every programming language, there is an associated software ecosystem that contains a large number of interdependent packages. These ecosystems provide valuable services to their respective software developer communities. For instance, the npm ecosystem has provided over

two million free and reusable software packages and has been trusted by more than 17 million developers all over the world (https://www.npmjs.com/, last accessed: 28th November 2023).

With increasing package counts (*e.g.*, forming a large tree of interdependent packages) within an application, the probability of incompatibility with dependencies increases. *Dietrich, Jezek & Brada (2014)* demonstrated that partial package upgrades can result in binary incompatibility during the build process. *Raemaekers, van Deursen & Visser (2017)* found that the inherent costs and risks of package incompatibilities make developers cautious when integrating new, unknown packages into their systems. *Kula et al. (2018)* reported that 69% of developers were unaware of the need to update packages and were not likely to prioritize an update due to its perceived additional labor requirements.

To address these challenges, developers have become increasingly dependent on various online platforms such as social media (*Guzman, Alkadhi & Seyff, 2017*; *Gonzalez-Marron, Mejia-Guzman & Enciso-Gonzalez, 2017*) and question-and-answer sites (*Meldrum, Licorish & Savarimuthu, 2017*; *Chakraborty et al., 2021*) since these networks have become mainstream communication channels. In a series of empirical studies, we examined the discussion related to the software ecosystem by developers on question-and-answer sites and found that package management issues are the biggest challenge (*Islam et al., 2021*, *2022*, *2023*). On the other hand, the software developers community heavily relies on Twitter's features for communication and sharing information, with noticeable variations in its usage among different groups (*Fang, Vasilescu & Herbsleb, 2023*). Besides its popularity, Twitter also offers developers to stay informed about fast-paced updates in the software development landscape, to learn from others, and to build relationships (*Sharma et al., 2018*; *Fang, Vasilescu & Herbsleb, 2023*). A thorough investigation of developers' tweets could be beneficial given the technical and social aspects leading to improved tool and process support in software development task (*Singer, Figueira Filho & Storey, 2014*; *Yasir et al., 2018*; *Sharma et al., 2018*). However, no study has been conducted that examines the tweets shared by developers to determine the challenges they face, including the topic of discussion, nature, and sentiments associated with the use of software ecosystems, while we do not explicitly compare the results with question-and-answer sites.

To bridge this gap, we empirically examine developers' tweets related to the software ecosystem in order to better understand how social media is used by them to deal with daily issues related to software development. The novelty of this study is that we are taking the first step towards understanding software ecosystem-related tweets that interest npm maintainers. Here, the term "npm maintainer" is defined as those developers who perform at least one pull request on the packages published in the npm ecosystem. We considered the npm ecosystem as a case study, since it is the largest software ecosystem used by developers worldwide and is also growing in popularity very rapidly (*Cogo, Oliva & Hassan, 2019*).

Our data collection method consists of three phases. First, we build npm maintainers dataset which results in 14,330 Twitter ID interlinked with GitHub. Second, we extract 39,425 tweets (*i.e.*, D1) posted by npm maintainers on the Twitter space. Third, we prepare a sample tweet dataset keeping 99% confidence with interval 3 which results in 1,176

```
TIL: npx is "an npm package runner" that comes pre-installed with #npm and
makes it easy to run locally installed npm CLI tools (e.g. `npm i -D parker`
and `npx parker`). This is useful if you  don't want to install a CLI tool
globally. https://t.co/0UIVcqfP5W
```

**Figure 1  A motivating example of a tweet posted by a npm maintainer related to software ecosystem.** The tweet topic relates to configuration, which falls under the package management category. Moreover, this tweet is posted inform of information with positive sentiment.

tweets (*i.e.*, D2). Figure 1 shows a motivating example of a tweet posted by a npm maintainer related to the software ecosystem. The topic of the tweet relates to configuration, which falls under the category of package management. Additionally, the tweet was posted as information with positive sentiment. To achieve more useful insight on tweets posted by npm maintainers, we qualitatively and empirically studied the topic of tweets ($RQ_1$), nature ($RQ_2$), and their sentiments ($RQ_3$).

According to the analysis results from $RQ_1$, a majority of tweets belong to the package management category, including topics such as usage scenario, configuration, feature information, bug report, software comparison, and dependency (58%). These findings indicate that Twitter is allegedly being used by developers and maintainers to filter and curate the vast amount of information related to package management activities in the npm ecosystem. The next popular tweet categories are notifications (28%) such as updates on a new release, followed by community-related information (4%). This information may prove useful to new developers when it comes to following npm maintainers on Twitter who are interested in specific aspects of software development. These findings align with our previous studies *Islam et al. (2021*, *2022*, *2023)* regarding package management issues of software ecosystem. From $RQ_2$, we find that npm maintainers who believe Twitter is an important tool for their development activities use a variety of strategies, with information tweets dominating, followed by question and answer. From $RQ_3$, we observe that npm maintainers post most tweets with neutral sentiments (51%), followed by positive (30%) and negative (13%). In detail, npm maintainers extensively utilize Twitter for package management in a neutral manner and community activities in a positive manner, which helps them maintain relationships with fellow developers, stay current on the latest software trends, and expand their knowledge of the software industry.

In sum, the contributions of this article are as follows:

- A novel dataset of 14,330 Twitter ID interlinked with the GitHub ID of npm maintainers and 39,425 npm ecosystem-related tweets posted by them.
- Identify the topic of tweets, nature and sentiment related to the npm ecosystem on Twitter for the first time. We conducted an open coding procedure on 1,176 tweets in order to categorize them and answer three research questions relevant to npm ecosystem discussions.
- A set of implications and recommendations for software developers, maintainers and researchers.

The rest of the article is organized as follows. The 'Related Work' section describes the related work. The 'Method' section presents our study settings to conduct the research. In detail, we explain the research questions, data collection process. The results of the study and their interpretations are described in the 'Results' section. The 'Implication and Recommendation' and the 'Threats to Validity' present the implication of our study and threats to validity, respectively. Finally, we conclude this article in the 'Conclusion and Future Works' section.

## RELATED WORK

In this section, we present the previous research works related to the analysis of software ecosystem.

### Studies on software ecosystem

A large number of interdependent packages, for example, increases the risk of incompatibilities within an application. Several recent studies have examined software ecosystem from the perspective of updates (*i.e.*, updating an existing dependency to a more recent version) and migrations (*i.e.*, replacing, removing, or adding a new package dependency) (*Cox et al., 2015*; *Bogart, Kästner & Herbsleb, 2015*; *Bogart et al., 2016*; *Decan, Mens & Constantinou, 2018*; *Abdalkareem et al., 2017*; *Kula et al., 2018*; *Decan, Mens & Grosjean, 2019*; *Cogo, Oliva & Hassan, 2021*; *Jafari et al., 2023*), security (*Zimmermann et al., 2019*; *Kabir et al., 2022*; *Wyss, De Carli & Davidson, 2022*; *Alfadel et al., 2023*) but the focus has not been on the package manager itself. According to *Bogart, Kästner & Herbsleb (2015)*, developers can make decisions about migration based on awareness mechanisms that incorporate various definitions of stability. *Bogart et al. (2016)* describes a number of reasons why developers fail to update, including community values, such as policies, support infrastructure, and accepted trade-offs to negotiate dependency changes. A study by *Jafari et al. (2023)* on how different characteristics of the npm package can influence the predicted update strategy reported that dependent count, age, and release status are the most influencing features. *Cogo, Oliva & Hassan (2021)* investigated npm dependency downgrades and found that the reasons for reactive downgrades were defects in a specific version of a provider, unexpected features of the provider, and incompatibilities. As reported in another study (*Kula et al., 2018*), 69% of developers were unaware of the need to update and were not likely to prioritize an update due to its perceived additional labor requirements. *Wyss, De Carli & Davidson (2022)* identified and characterized 6,292 shrink wrapped clones and found that up to 2,159 depended on outdated and vulnerable npm package dependencies. Furthermore, 207 clones contained vulnerabilities that were not detected by npm audits. *Abdalkareem et al. (2017)* examined the reasons why developers use trivial packages in the npm ecosystem and found that developers consider these packages to be well tested and well implemented, making them more productive because they do not need to implement small features. *Zimmermann et al. (2019)* conducted a case study on npm security and found that there are a number of single points of failure with npm and that unmaintained packages pose a threat to large code bases. These studies have demonstrated that developers have difficulties in managing dependent packages.

## Studies on communication platforms

Software developers now rely heavily on different communication platforms such as Twitter and question-and-answer sites such as Stack Overflow (*Meldrum, Licorish & Savarimuthu, 2017*; *Chakraborty et al., 2021*) or a project-specific forum such as Eclipse forums (*Nugroho et al., 2021*) and GitHub discussion (*Hata et al., 2022*) to coordinate communication and learning, share knowledge, and recruiting activities (*Meldrum, Licorish & Savarimuthu, 2017*; *Begel, DeLine & Zimmermann, 2010*). Several empirical studies were conducted on Twitter posts to explore software engineering trends (*Guzman, Alkadhi & Seyff, 2017*; *Gonzalez-Marron, Mejia-Guzman & Enciso-Gonzalez, 2017*; *Bougie et al., 2011*; *Fang et al., 2020*), recommendation tools support (*Sharma et al., 2018*), community detection (*Lim & Datta, 2012*) *etc*. These prior studies show that Twitter posts are useful in understanding software developers' challenges. In our previous empirical studies, we looked at software ecosystem-related discussions by developers on question-and-answer sites and found that package management issues are the biggest challenge (*Islam et al., 2021*, *2023*, *2022*). However, no study has been conducted so far to examine the tweets shared by developers to reveal the challenges associated with using the software ecosystem and how they differ from question-and-answer sites.

In this study, we empirically examine developers' (*i.e.*, npm maintainers) tweets related to the software ecosystem in order to better understand how social media is used by them to deal with daily issues related to software development.

## METHOD

The purpose of this research is to investigate the software ecosystem related to tweets by npm maintainers. We selected npm as a case study, as it is the largest software ecosystem trusted by more than 11 million developers around the world (https://zenodo.org/record/8246509). Figure 2 shows the overview of the methodology of our study.

In detail, we present the data set used in this work, describing the rationale for this particular choice, along with the procedure performed to collect the dataset from the primary sources. Furthermore, we state that all the methods were carried out in accordance with the relevant GitHub and Twitter guidelines and regulations. More details can be found in the Supplemental Information. For the data collection method, please visit the following link: https://zenodo.org/record/8246509.

### Research questions

To guide the study, we formulate the following research questions with their motivations.

- *Topic of tweet (RQ₁): What are the main topics that npm maintainers communicate on Twitter? Motivation:* The npm maintainers are key decision-makers in software project development. Hence it is important to understand how they leverage Twitter as a medium.

- *Nature of tweet (RQ₂): What is the formulation nature of tweets communicated by npm maintainers on various topics? Motivation:* Having identified the topics of interest, it is important to know the nature of discussions that happen in the context of these topics,

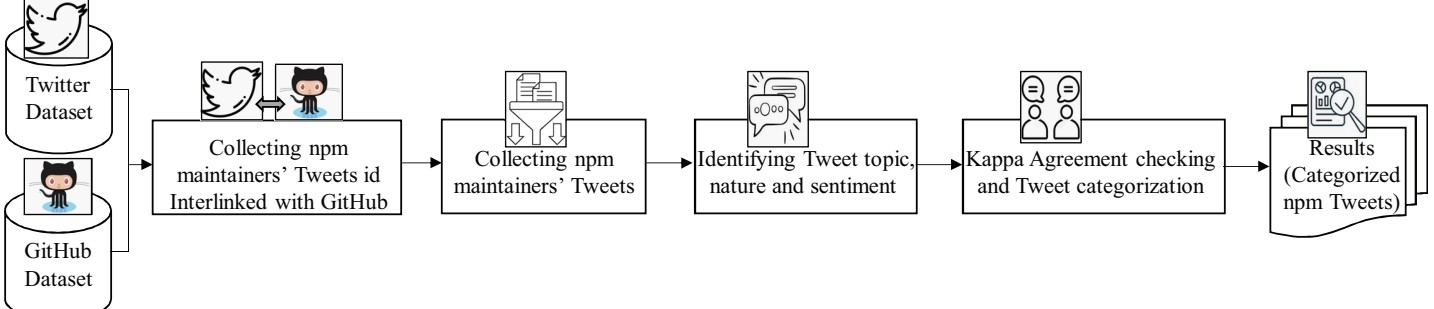

**Figure 2 Overview of the methodology of our empirical study.** The data collection process consists of three steps: (i) Building npm maintainers Twitter ID dataset and (ii) Extracting npm maintainers tweets, and (iii) preparing a sample tweet dataset. Afterward, we comprehensively analyze the nature of npm maintainers' communication through Twitter regarding the topics, nature, and sentiments.

where npm maintainers posts are replied to by others (*Guzzi et al., 2013*; *Yasir et al., 2018*). Furthermore, Twitter is an informal communication channel that is not limited to discussing problems and solutions, but also allows developers to post questions on information tweets, which differs from Q&A forums, such as Stack Overflow.

- *Sentiment of tweet (RQ$_3$): What is the sentiment of tweets by npm maintainers on Twitter? Motivation:* The motivation of RQ$_3$ is to get deep insight into developers' sentiments while sharing npm package ecosystem-related information. *Stieglitz & Dang-Xuan (2013)* reported that tweet sentiment is an important factor in the diffusion of information on Twitter. *Tourani, Jiang & Adams (2014)* adopted sentiment analysis technique to identify distress or happiness in a software development team. In this RQ, we aim to test the degree of satisfaction/dissatisfaction while sharing npm-ecosystem related tweets.

## Data collection

In this section, we describe our methods for collecting npm ecosystem-related tweets by software developers. The data collection process consists of three steps: (i) Building npm maintainers Twitter ID dataset and (ii) extracting npm maintainers Tweets, and (iii) preparing a sample tweet dataset. Details of the data collection procedure are explained below:

- *Step-1: Building npm maintainers Twitter ID dataset-* For an in-depth examination of npm ecosystem-related issues, we primarily focused on npm maintainers tweets. To identify npm maintainers, we only select those who performed at least one pull request on software packages published on npm ecosystem. To do so, first, we extracted all npm pull requests through the GitHub API and obtained 123,647 npm maintainers GitHub ID. We subsequently extracted user profile information for each GitHub ID. The output of this step is 14,330 Twitter IDs of npm maintainers interlinked with GitHub IDs.

- *Step-2: Extracting npm maintainers tweets-* To extract npm maintainers Tweets using the output of step-1, we utilize the official Twitter search API. Afterwards, to filter npm ecosystem-related Tweets we limit our data collection process to tweets that only correspond to specific keywords such as npm, pnpm, npm-install, npm-scripts, npmignore, npm-shrinkwrap obtained from our previous work (*Islam et al., 2023*). While extracting the data, we found many non-English tweets in the dataset. However, we decided to keep them in the dataset for a couple of reasons. First, we aimed to maintain the dataset's integrity and inclusiveness, reflecting the actual data we collected during the time of the study. Second, preserving these non-English tweets allows for the potential exploration of cross-language sentiments or topics in future research. The output of this step is 39,425 tweets dataset (D1) related to npm ecosystem by npm maintainers.

- *Step-3: Preparing sample tweet dataset-* After obtaining the npm ecosystem-related tweets (*i.e.*, dataset D1) posted by npm maintainers, we prepare a sample dataset keeping 99% confidence with interval 3 (*Aghajani et al., 2019*; *AlOmar et al., 2022*). The output of this step is 1,176 tweets dataset (D2).

## Approach

In this section, we describe the approach we have taken to answer each research question. We highlight that our study's focus is on qualitative coding to capture topics, nature, and the sentiment of tweets. Qualitative coding provides a more detailed and context-rich understanding, especially considering the inherent complexities of the text in our dataset, such as sarcastic tweets. This methodology allowed us to capture subtle variations in communication that automated techniques (*e.g.* topic modeling) might miss. The detailed approaches of each research question are outlined below:

### Topic of tweet (RQ₁): what are the main topics that npm maintainers communicate on Twitter?

To answer $RQ_1$, we conducted a qualitative analysis of statistically representative samples included in dataset D2. In order to identify the topics discussed by npm maintainers on Twitter, we manually analyzed the tweets related to npm ecosystem using thematic analysis similar to previous studies (*Hata et al., 2019*; *Nugroho et al., 2021*, *2022*; *Islam et al., 2023*). This process consists of four steps: (1) building a schema to classify the tweets, (2) grouping the topics into coherent categories, (3) classifying the tweets manually based on the developed schema, (4) aggregating the results from analysis. Here, the term "tweets categories" is used to represent the main themes or subjects of the tweets, "topic of tweets" refers to the specific classes or groups into which the tweets are classified.

In step-1, our initial set of topics was determined by preliminary analyses of 30 random tweet sample dataset, as well as by borrowing useful categories from previous related research works (*Sharma, Tian & Lo, 2015*; *Guzman, Alkadhi & Seyff, 2017*; *Yasir et al., 2018*). This step generates an initial list of 20 npm ecosystem-related topics discussed by npm maintainers on Twitter. In step-2, the initial topics were then refined through

collaborative discussions between two authors (*i.e.*, 1st and 2nd authors). The tweet topic refining process consisted of two phases. Firstly, the 1st and 2nd authors conducted a manual analysis of 30 random sample tweets and compared their agreement levels. To resolve disagreements in results, both the 1st and 2nd authors discussed and agreed on a suitable code for the respective tweet. In this phase, the topics were revised and merged into coherent categories. In this process, 19 tweet topics were generated under five coherent categories. Following this, the same two authors manually annotated 30 additional samples to ensure that no new topics had emerged. This step resulted in the same 19 tweet topics being categorized into five coherent categories. In Table 1, the 19 tweet topics are summarized and their definition with representative examples are provided below:

- *Media sharing:* Sharing articles, blogs, or tutorials, related to npm package ecosystem. For example: "Slides for my recent @reactdelhincr session React 16 and; NPM, Create your own library" can be found here—https://t.co/8RRLoYEDiP #reactdelhincr #reactjs #npm #javascript #OpenSource".
- *New release/progress update:* Updates about developing a module/feature/method/class of npm package. For example: "I published my first official open source package on npm today. A simple, accessible autocomplete component for vanilla JavaScript and Vue. https://t.co/1O9LvjUQtY".
- *General news:* About general news relating to the software industry. For example: "Microsoft buying npm is such a great move, I can't wait to see deeper integrations with github".
- *Product promotion:* Promotion of commercial books or tools related to software development. For example: "#npmjs will reach a million packages within a couple of months. That's a lot to choose from".
- *Career:* Related to job openings and candidates sharing their availability for hire. For example: "@onel0p3z RT @seldo: So @npmjs is gonna be hiring an ops person in next week or twoz".
- *Usage scenario:* Explanation or discussion about the use of the npm package ecosystem. For example: "I think in the modern JS ecosystem, we should emphasize writing isomorphic code. It's a shame how many npm packages there are where the core functionality could come in handy in the browser, but you can't use it because it depends on "fs".
- *Software comparison:* npm's working is being compared against any other software affiliated or not affiliated with npm or the integration with other software. For example: "Also smh was trying to use yarn solely for it's less verbose package installing and then npm notice" would always add a new package-lock.json even though there are yarn.lock files spread all over… eventually buckled in to just use npm but bleh".
- *Dependency:* Discussion about the npm package dependency management in a project. For example: "@zkat Hm, well currently I am working on a linter for your

**Table 1  Identified topic of tweets by npm maintainers.**

| Category | Topic labels | Label source |
|---|---|---|
| Notification | Media sharing | *Sharma, Tian & Lo (2015)* |
| | New release/update | *Sharma, Tian & Lo (2015)* |
| | General news | *Sharma, Tian & Lo (2015)* |
| | Product promotion | *Sharma, Tian & Lo (2015)* |
| | Career | *Sharma, Tian & Lo (2015)* |
| | Usage scenario | *Yasir et al. (2018)* |
| | Software comparison | Our work |
| Package management | Dependency | Our work |
| | Configuration | Our work |
| | Feature info | *Guzman, Alkadhi & Seyff (2017)* |
| | Feature request | *Guzman, Alkadhi & Seyff (2017)* |
| | Bug report | *Guzman, Alkadhi & Seyff (2017)* |
| | Build | Our work |
| | Crowdsourcing request | *Sharma, Tian & Lo (2015)* |
| Community related | Community event | *Yasir et al. (2018)* |
| | Community info | *Yasir et al. (2018)* |
| | Personal promotion | *Yasir et al. (2018)* |
| | Satire | *Guzman, Alkadhi & Seyff (2017)* |
| Other | Not in English | *Yasir et al. (2018)* |
| | Other | *Guzman, Alkadhi & Seyff (2017)* |

**Note:**
In summary, we identified 19 topics from npm maintainers' tweets related to software ecosystem and categorized them into five major categories including others.

dependencies". Because npm is written in js, I can just import those libs and build on top. How might I go about doing this type of project on Orogene?".

- *Configuration:* Discussion about managing configuration (*i.e.*, simple installation, environment setting, *etc*) of npm package ecosystem. For example: "@stevensanderson You could use the benchmark harness in @stdlibjs (see https://t.co/XlGPUicd3x). One can bundle as a standalone file to use independently of stdlib. More work than 'npm install', but doable. We use it extensively (*e.g.*, https://t.co/e7GReb)".

- *Feature information:* Mentioning a specific feature without any objective evaluation. For example: "@alexanderKaran Thanks, just tried, looks neat, just one problem. The default command 'ncu -g' does not come with the URL of the repo like npm-check, would be handy to include this for checking the release note. https://t.co/HjfLbbI8dZ".

- *Feature request:* An explicit request for an update on an existing feature or new specific feature to be developed. For example: "@jeffbcross @getDanArias @angular @NxDevTools Be great if a library can be ejected out as an independent npm module".

- *Bug report:* Report of an error, flaw, failure or fault for existing features. For example: "@code—the auto update paths feature has a bug. This happens when I copy+paste a dir in the sidebar, and then rename a file in the copied dir. Each open file has npm imports changed to rel typescript imports (I had "javascript.updateImportsOnFileMove.ena")".

- *Build:* Talks about compiling software/code/make file/build failure in the system. For example: "@isntitvacant So useful for executing one off build scripts (*e.g.*, npx webpack; npx electron.)! @npmjs".
- *Crowd sourcing request:* Requesting community to contribute to open source projects, surveys, petitions, *etc.* For example: "@JessTelford @npmjs @yarnpkg Personally I would say open source would make me at a minimum 1% more efficient. I wonder if we all donated even 0.5% of what people earn to this sort of solution what it would look like. Approx $42 a month (based on 100k sala)".
- *Community event:* About community events (Conferences, coding events, anniversaries, *etc.*). For example: "Added Node.js Interactive Europe to the Awesome Tech Conferences curated list! https://t.co/IL2oFfiNhs #nodejs #javascript #npmjs".
- *Community information:* Raising awareness of an issue/info relevant to the npm community. For example: "TIL: @npmjs sponsors a wombat called Teacup and they have a dedicated Slack channel where they put updates on her growth along with pictures https://t.co/AJYI4sYPAR".
- *Personal promotion:* Promotion of an individual's social activities (talks, workshops, and write-ups) in events which are relevant to the community. For example: "Shout out to @sindresorhus for all his gems on @npmjs Creating a few CLIs and they are littered with his work. Thanks to the #cliNinja".
- *Satire:* Humorous content from npm maintainers. For example: "Today I learned that 'npm' does not stand for Node Package Manager: https://t.co/J2dQnfCE".
- *Not in English:* Tweets with more than one word not written in English. For example: "@12120121201201 @DamianCatanzaro Seguro ya hay una lib deprecada en npm que hace eso".
- *Other:* Tweet is not covered by existing categories. For example: "@Kosai106 Is he the one who wants your npm namespace?" or "Sunday result: #emacs #nodejs #expressjs #redis #socketio #jade #jquery #npm #tor #torsocks".

In step-3, a shared understanding was established, followed by several rounds of coding in the same general manner as in step-2. In contrast, step-3 was intended to determine the reliability between the raters (*i.e.*, the third and fourth authors). According to our guidelines for annotations, we did not allow multiple categories for one tweet. This step results in a satisfactory kappa score (*Viera & Garrett, 2005*) (*i.e.*, {Free marginal kappa = 0.79, Fixed-marginal kappa = 0.76} for tweet topic and {Free-marginal kappa = 0.87, Fixed-marginal kappa = 0.86} for tweet categories) with no new tweet topic proposal. Based on the inter-reliability between the raters and the annotation guidelines, the remaining samples were manually annotated. In step-4, we analyzed the distribution of tweet topics and their coherent categories discussed by npm maintainers.

### Nature of tweet (RQ$_2$): what is the formulation nature of tweets communicated by npm maintainers on various topics?

To answer RQ$_2$ (*i.e.*, understanding the nature and purpose of the tweets posted by the npm maintainers), we qualitatively coded the statistically representative sample tweet dataset D2 using contextual information from the tweet conversation. The entire analysis process is accomplished in four distinct steps. In Step 1, we build a coding schema by borrowing useful categories from a previous study (*Fang et al., 2020*; *Yasir et al., 2018*). In Step 2, we refined the coding schema through collaborative discussion between two authors (*i.e.*, the first and second author) similar to RQ$_1$. This step resulted in four categories of tweet nature, including others. Details of the tweet nature classification scheme are described below:

- *Question:* Tweets in this nature ask about technical details of the npm ecosystem, or they refer to an open issue. For example, "@clemp6r Thank you buddy! Was the fork; npm install enough to get started?" is a tweet asking for technical details about npm installation-related issue.

- *Answer:* These tweets respond to questions posed by other Twitter users. They may provide technical details regarding the use of the npm ecosystem, or refer visitors to other resources that are useful in solving problems. For example, "@joezo it's bug from atom version of npm" is an answer type tweet in response to a question on atom version of npm.

- *Information:* Tweets that provide information about the npm ecosystem, a file, an issue discussion, or other work-related news, without directly advocating its use. This category includes tweets that are not part of a Twitter conversation. For example, "Every package manager should support a home command like npm. I just hacked a quick script to easily open the homepage of opam packages in Chrome. https://t.co/q6hnOflnuf".

- *Other:* If the tweet characteristics do not fit the above classes. For instance: "Sunday result: #emacs #nodejs #expressjs #redis #socketio #jade #jquery #npm #tor #torsocks".

In Step 3, we build a shared understanding among authors through several rounds of manual annotation of sample tweets. As part of our annotation guidelines, multiple categories were not allowed for a single tweet. To verify the quality of our classification, we performed a kappa agreement check using 30 random samples among two authors. Using the kappa score calculator, we evaluated the degree of agreement and concluded that the overall score was satisfactory (*i.e.*, Free-marginal kappa = 0.87, Fixed-marginal kappa = 0.85) (*Viera & Garrett, 2005*). After obtaining the consent of both authors, the remaining sample tweets were manually annotated by them. In Step 4, we analyzed the results from npm maintainers' conversations on Twitter related to different topics in the form of questions and answers.

***Sentiment of tweet (RQ₃): what is the sentiment of tweets by npm maintainers on Twitter?***

To answer RQ₃, we performed qualitative analysis on the statistically representative sample of tweets included in dataset D2. We used the same coding scheme used by previous studies (*Williams & Mahmoud, 2017*; *Nugroho et al., 2021*) to assess sentiment in the npm maintainers' tweets. Details of the tweet coding scheme are described below:

- *Positive:* It includes tweets that inspire positive emotions while reading them and also mentions positive words (*e.g.*, nice, good, thanks, *etc.*). For example: "@zeroload @npmjs Nice. Instant is definitely nice. But I think surfacing good libraries is the part that needs more desperate solving" is a positive sentiment tweet.
- *Negative:* It includes tweets that cause negative emotions while reading them and also mention negative words (*i.e.*, failure, unfortunately, bug, *etc.*). For example: "Do not upgrade to node 0.10.19! npm install often fails with npm ERR! cb() never called!" GitHub Issue: https://t.co/vwyUzjSOPV" is a negative sentiment tweet.
- *Neutral:* It includes tweets that have neutral feelings (*e.g.*, those that began with positive speech and ended with negative speech) as well as those that do not include biased words. For example: "@makeusabrew Hi Nick! Are you by any chance planning to use that npm package name 'angle'? Asking to use it for https://t.co/ulhHmD2TZT." is a neutral sentiment tweet.
- *Other:* If the sentiment of the tweet does not eligible to classify to the other three sentiments. For instance: "Sunday result: #emacs #nodejs #expressjs #redis #socketio #jade #jquery #npm #tor #torsocks".

As part of our annotation guidelines, we did not allow multiple categories for a single tweet. We conducted a Kappa agreement check on 30 random samples among two authors in order to verify the quality of our classification. Based on the Kappa score calculator (*Viera & Garrett, 2005*), we evaluated the degree of agreement and concluded that the overall score is 'satisfactory' (*i.e.*, Free-marginal kappa = 0.78, Fixed-marginal kappa = 0.72). Based on this agreement between the two authors, the remaining sample tweets were then manually annotated by the two authors.

## RESULTS

In this study, we aim to comprehensively understand the nature of npm maintainers' communication through Twitter regarding the topics, nature, and sentiments. In detail, we describe the findings of this study per research question as follows.

### Topic of tweet (RQ₁): what are the main topics that npm maintainers communicate on Twitter?

Figures 3 and 4 illustrate the results of our qualitative analysis of the tweets posted by the npm maintainers. As a whole, tweets belonging to the package management category, including topics such as usage scenarios, configurations, feature information, bug reports, software comparisons, and dependencies, were most prevalent (*i.e.*, 58%). This finding is

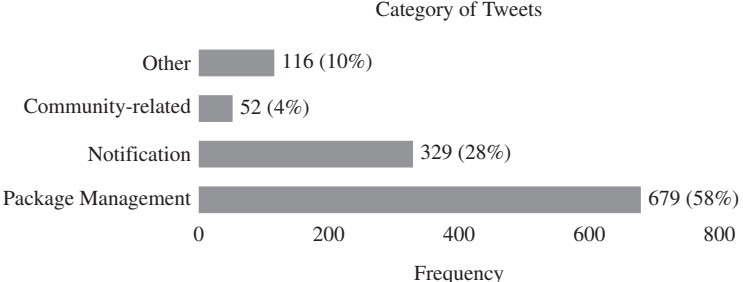

Category of Tweets

**Figure 3 Distribution of tweet categories across 1,176 tweets.** Of the four identified categories, 'package management' and 'notification' are the most categories discussed by npm maintainers on Twitter.               

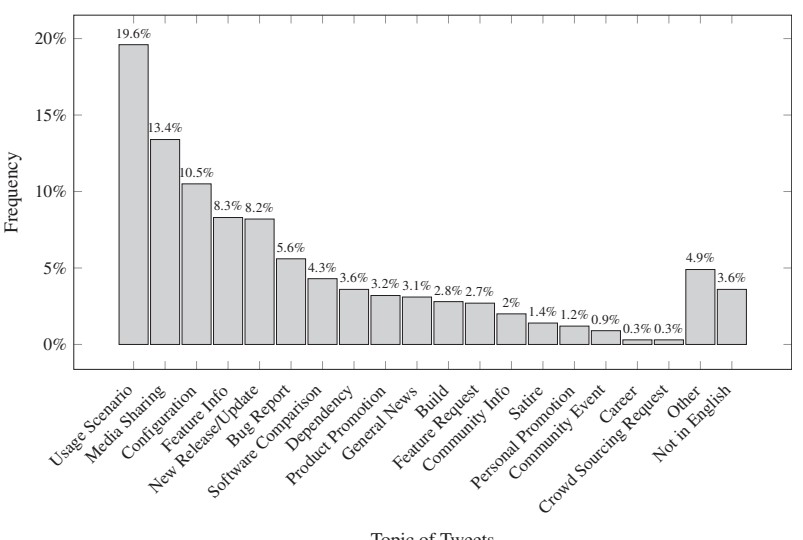

**Figure 4 Frequency of each topic amongst 1,176 tweets by npm maintainers.** In line with Fig. 3, the 'usage scenario', which belongs to the 'package management' category, is the most dominant topic communicated in Twitter space.               

consistent with our previous studies on software ecosystem issues (*Islam et al., 2021*, *2023*). The findings indicate that developers and maintainers are using Twitter to filter and curate information on package management activities associated with the npm ecosystem. Following package management issues, tweets belonging to the notification category, including topics such as media sharing, new releases/progress updates, product promotions, and general news, were the most common (28%). In light of this finding, it appears that npm maintainers utilize Twitter to stay informed, maintain relationships with developers, and keep up to date with their domain knowledge. Finally, tweets that relate to the communication category include topics such as community information and community events, which are the most prevalent (4%). Even though community-related tweets are relatively low in number, the large number of tweets communicated every day for the applications in our data sample suggests the number of relevant tweets is significant, and should therefore be considered by software companies when preparing the next version of their applications. *Singer et al. (2013)* also reported that it is common for

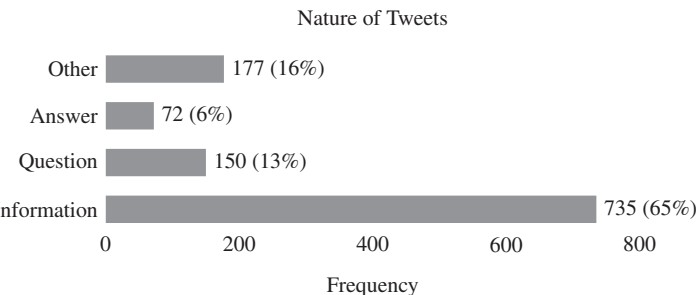

**Figure 5 Distribution of the nature of 1,134 tweets.** We found three main nature of tweets after excluding the non-English tweets, in which 'information' tweets are the most dominant type posted by npm maintainers.

developers to use social media to communicate and collaborate with one another. Therefore, npm maintainers should pay attention to update their information in the software ecosystem community.

### Nature of tweet (RQ₂): what is the formulation nature of tweets communicated by npm maintainers on various topics?

Figures 5–7 illustrate the results of our qualitative analysis to identify the nature of tweets. We observe that npm maintainers who feel that Twitter benefits them rely on a variety of strategies for posting content related to software ecosystem. In Fig. 5, developers post npm ecosystem-related tweets in form of information (65%) followed by question (13%) and answer (6%). In contrast of tweet categories *vs* nature as shown in Fig. 6, we find that notification (79.9%) related tweets are mostly shared in the form of information followed by package management (65%), and community-related issues (48.1%). In addition, we find that 18.7% tweets in package management-related issues are posted in the form of questions by developers. In contrast to tweet topic *vs* nature as shown in Fig. 7, we find that usage scenario (59.7%), software comparison (66%), configuration (71%), build (75.8%), feature information (80.6%) related tweets are dominantly shared in form of information. In addition, question-type tweets are significant in a bug report (25.8%), usage scenario (22.4%), feature request (21.9%), *etc*. This finding aligns with our previous study (*Islam et al., 2022*). It appears that npm maintainers use Twitter more for technical issues (such as package management) than for community-centric events and activities. It is important to note that while social media platforms claim to predominantly support community-related activities, we observe a predominance of technical work-oriented discussions in the form of information. In summary, tweets regarding the software ecosystem concerning package management work have gained greater attention in terms of the volume of information provided.

### Sentiment of tweet (RQ₃): what is the sentiment of tweets by npm maintainers on Twitter?

Figure 8, and nine illustrate the results of qualitative analysis to identify the sentiment of tweets. Overall, npm maintainers mostly post tweets with a neutral sentiment (51%) followed by positive (30%) and negative sentiment (13%) as shown in Fig. 8. It is possible

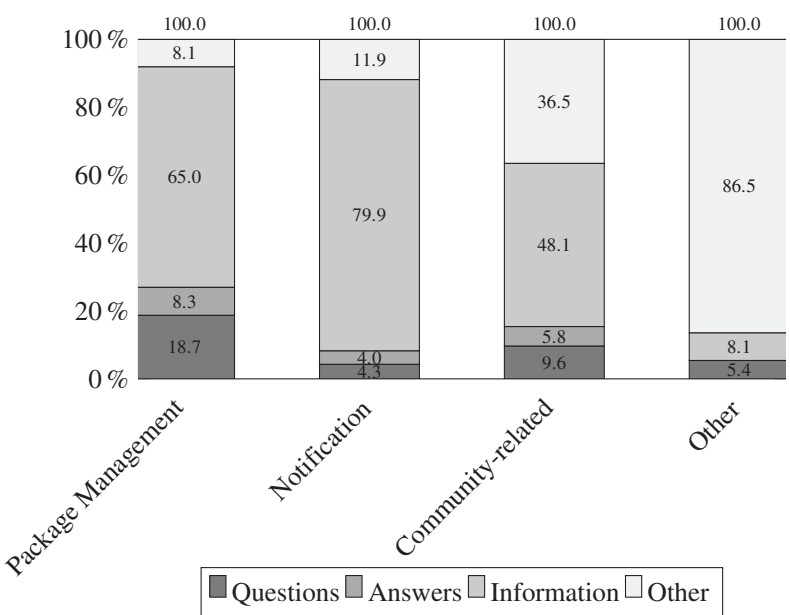

**Figure 6 Distribution of tweets categories *vs.* nature amongst 1,134 tweets.** Most of the tweets related to notifications are shared as information, followed by tweets related to package management and community-related issues.

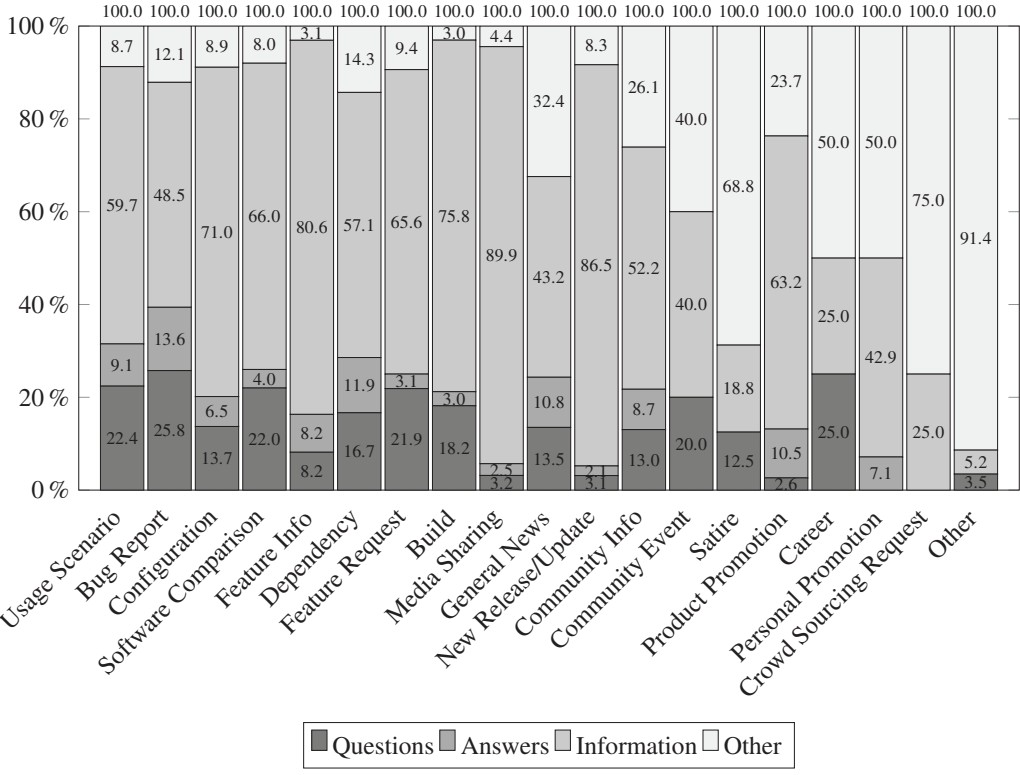

**Figure 7 Distribution of tweets topics *vs.* characteristics amongst 1,134 tweets.** The most frequently shared information on Twitter is regarding usage scenarios, software comparisons, configurations, builds, and feature information.

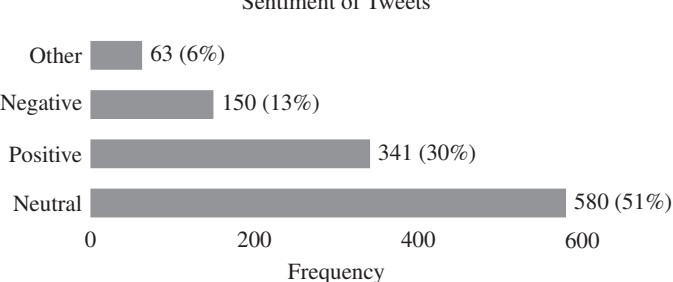

**Figure 8 Distribution of sentiment across 1,134 tweets after excluding non-English tweets.** In general, npm maintainers mostly tweet in a neutral tone, followed by positive and negative tones.

that a large number of tweets with a neutral sentiment can be attributed to the proliferation of tweets from senior developers. In contrast of tweet categories *vs* sentiment as shown in Fig. 9, we observe that package management-related tweets are mostly neutral (61%) followed by positive (30.5%) and negative (17.5%) sentiment. This hints that npm maintainers have better experience in package management when using npm ecosystem. In the notification category, most tweets belong to positive sentiment (51.1%) followed by neutral (43.5%) and negative (5.5%). In the community category, most tweets belong to positive (57.7%) sentiment followed by neutral (23.1%) and negative (19.2%) sentiment. The results indicate that npm maintainers extendably use Twitter for community activities in a positive manner to stay up-to-date with the latest software trends and practices, to expand their software knowledge by learning new stuff, and to maintain relationships with fellow software developers.

## IMPLICATION AND RECOMMENDATION

In this section, we present a description of the impact of our study results and recommendations, as follows:

### Implications

Twitter provides developers and package maintainers with a means of staying up to date with software engineering trends and practices, facilitating their learning, and building relationships among themselves. We investigated how the maintainers community use Twitter in order to support their npm development activities and the challenges associated with using software ecosystems. The results of our study demonstrate that Twitter contains useful information for the npm developers and maintainers community. Based on the results of our study, we provide a list of implications based on the analysis results:

- According to our analysis, the majority of tweets by npm maintainer are related to package management issues, such as usage scenario, configuration, feature information, bug, software comparison, and dependency, *etc*. This result aligns well with our previous works (*Islam et al., 2021*, *2023*) using Stack Overflow, which also reported that developers struggle mainly with package management issues. The findings indicate that developers and maintainers use Twitter to filter and curate information related to

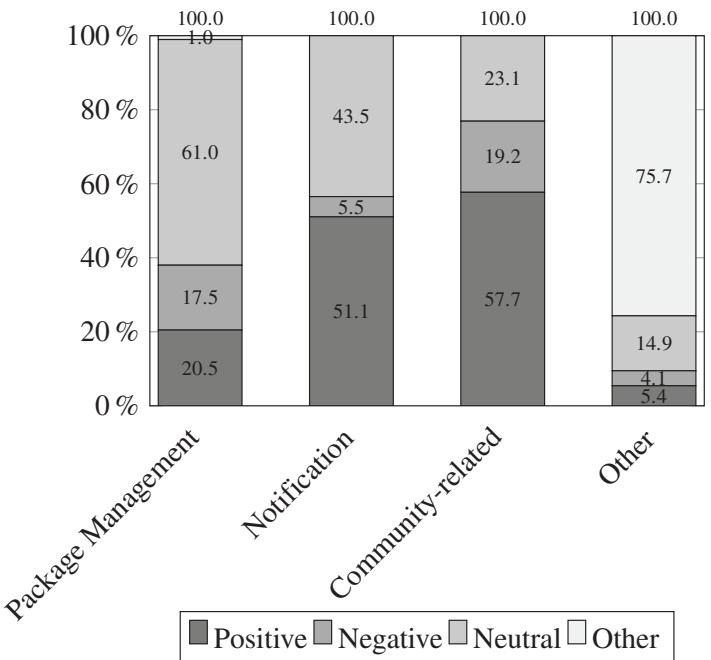

**Figure 9 Distribution of sentiment-based tweets categories amongst 1,134 tweets.** Most package management-related tweets have a neutral tone, followed by positive tweets and negative tweets. In addition, maintainers use Twitter extensively for community activities in a positive manner to stay up-to-date with the latest software trends and practices, to expand their software knowledge by learning new things, and to maintain relationships with other software maintainers.

package management issues in the npm ecosystem. Therefore, developers and maintainers should consider informal communication channels such as Twitter as well as formal channels in order to gain a more comprehensive understanding of package management issues. The next most popular category of tweets is notifications, such as updates about a new release, followed by community-related information. This finding is consistent with previous studies (*Singer, Figueira Filho & Storey, 2014*; *Tian & Lo, 2014*). In detail, a survey by *Singer, Figueira Filho & Storey (2014)* reported that many developers use Twitter to stay up-to-date with the latest trends, consume knowledge, and network with each other. Another study by *Tian & Lo (2014)*, found that some software developers are very active in posting software-related content. This implies that informal communication channels such as Twitter are adopted by developers and maintainers as an easy way to reach the largest community in the npm ecosystem since the formal channel offers limited features to developers (*Yasir et al., 2018*). Hence, if npm maintainers plan to inform any information related to the software ecosystem, they should keep this in mind. Moreover, a developer who is just beginning to engage in social media should consider adopting the above strategies in order to increase their chances of staying informed, learning, and developing relationships. For instance, follow the most active npm maintainer tweets in your field, in order to obtain verified information on package management issues. However, choosing whom you follow

incurs a cost in terms of attention; therefore, be careful in your selection of whom to follow.

- Developers and package maintainers who consider Twitter an important tool for their development activities use a variety of strategies to post content on. Information-type tweets are the most common format for posting content on Twitter followed by question and answer. This finding aligns well with previous studies (*Hughes & Palen, 2009*; *Tian et al., 2012*; *Yasir et al., 2018*). Our study results show that npm maintainers prefer to share tweets containing information concerning what they learn about a new technology or related practices. This implies that Twitter helps developers and maintainers stay aware of package management tools and practices, learning support, and plays a role in building a collaborative community through sharing useful information related to npm ecosystem. In addition, developing a recommendation system capable of identifying tweets that would be of interest to many developers (*e.g.*, tweets about package management) in order to help them remain current with recent development and gain new knowledge.

- Sentiment analysis revealed that the majority of tweets about the npm ecosystem were neutral, with a smaller proportion expressing positive or negative sentiment. This finding align well with prior study by *Guzman, Alkadhi & Seyff (2017)*. They reported that the overall sentiment polarity of tweets related to software applications is neutral. This suggests that developers generally have a neutral attitude towards the npm ecosystem, with some expressing enthusiasm or dissatisfaction. As a result of these findings, we can infer how typical development circumstances (such as review of package management features, usage scenarios, *etc.*,) influence the sentiments of members of a software development team (*Tourani, Jiang & Adams, 2014*; *Bano & Zowghi, 2015*; *Lin et al., 2018*). Furthermore, we can identify how sentiments are spread within the npm maintainers community and how measures and methods can be implemented to minimize negative effects and enhance positive aspects that users and developers will perceive positively. For instance, such analyses can be utilized to understand and rationalize developers' and package maintainers sentiments toward newly released packages and features, helping software developers plan better for future releases.

## Recommendations

Besides the results of our study have shown some practical implications in the software engineering area, we also provide a set of recommendations for developers, package maintainers, and researchers, as follows:

- Package maintainers and developers should prioritize the usage scenario of a package to ensure a positive experience for new users.
- The npm ecosystem should continue to monitor and address any negative sentiment expressed by package maintainers to improve user satisfaction.

- Further research could be conducted to identify the specific issues or features that are causing negative sentiment among developers, in order to address them more effectively.
- The findings of this study also recommend that social media posts, such as Twitter posts, should be considered in conjunction with traditional question-and-answering websites, such as Stack Overflow when evaluating technical discussions relating to software ecosystems.

## THREATS TO VALIDITY

This section describes the internal, external, and construct validity threats of this study.

### Internal validity

Threats to internal validity relate to experimental bias and error in conducting the analysis. The first threat is the accuracy of the methods used in this study. We perform manual analysis on a random sample since the dataset size is large. To mitigate this challenge, we prepare representative samples for the tweet dataset, with a confidence level of 99% and an interval of three. Thus, we believe that experimental bias and error in conducting the analysis were reduced. Another internal validity is regarding the manual analysis that may produce the potential subjectivity of the study. However, we intentionally chose this approach for several reasons. Manual analysis not only helps us better understand the sentiment nuances, but also allows us to gain insights into the contextual intricacies that contribute to npm maintainers perceptions. In addition, several prior research works (*Williams & Mahmoud, 2017*; *Nugroho et al., 2021*) also utilize manual classification approach to understand different community issues due to domain-specific meanings and expressions in text.

### External validity

Threats to external validity relate to the generalizability of findings. In our study, we focus only on Twitter which is one of the largest and most popular social media platforms among end-users. In the context of identifying npm maintainers, it is essential to note that the requirement to link a Twitter account to a GitHub profile may result in the inadvertent exclusion of developers who do not have their Twitter accounts linked. Thus, the results may not represent the other npm maintainers who do not link their Twitter accounts to GitHub. Although we conducted a large-scale study of npm-related discussions on Twitter, the findings may not generalize to other question-and-answer sites and social media platforms. However, our study is consistent with previous works that also utilized Twitter data (*Bougie et al., 2011*; *Guzman, Alkadhi & Seyff, 2017*; *Fang et al., 2020*; *Gonzalez-Marron, Mejia-Guzman & Enciso-Gonzalez, 2017*).

### Construct validity

Threats to construct validity are related to potential errors that can occur when extracting data about npm ecosystem-related discussions. The first construct validity relates to the data collection. We used keywords to identify posts related to the npm ecosystem, but

some posts may be incorrectly labelled. To reduce this threat, we created the list of keywords from our previous published article (*Islam et al., 2023*).

In the qualitative analysis of classifying Tweet types, the posts may be mislabelled due to the subjective nature of our coding approach. Despite annotators resolving disagreements through discussion, the labels might still be incorrect. There are also many potential factors that need to be taken into account. To mitigate this threat, we limit the scope of our study to conversational contents of the discussions themselves, to gain insights into how Twitter is utilized. In addition, we took a systematic approach to validate the taxonomy and the comprehension understanding among the first four authors of this article in several rounds. Only when the Kappa score reaches an acceptable range and thus we are able to complete the rest of the sample dataset.

## CONCLUSION AND FUTURE WORKS

In this article, we empirically examine npm maintainers' tweets related to software ecosystem in order to better understand how social media is used by them to deal with daily issues related to software maintenance activities. We collected a dataset of tweets and analyzed them qualitatively in order to identify their topics, nature, and sentiments. We found that the majority of tweets belong to the package management category, followed by notifications and community-related tweets. The most frequently discussed topics among npm maintainers in the package management category are usage scenarios. We observed that the nature of tweets mostly shared by npm maintainers is information, followed by question and answer, respectively. Furthermore, sentiment analysis revealed that developers express a more positive sentiment towards notification and community-related discussions, whereas they express a more neutral sentiment towards discussions pertaining to package management. In conclusion, npm maintainers extensively utilize Twitter as part of their package management and community-centered activities to stay up to date on the latest software trends and practices, expand their knowledge of the software industry, and maintain relationships with other software developers.

This study will be extended in the future to include tweets about other software ecosystems and compare how npm maintainers discuss issues to better understand noteworthy tweets such as security-related tweets in the Twitter space. In addition, we can also encompass diverse aspects such as maintainers' interactions with specific npm packages, sentiment variations based on involvement levels, correlations between package popularity/quality and tweet content, and the usefulness of different types of tweets. These are all compelling areas that can contribute to a more comprehensive understanding of the dynamics within the developer and maintainers community on Twitter. Finally, we intend to develop a recommendation system capable of identifying tweets that would be of interest to many developers (*e.g.*, tweets about package management) in order to help developers remain current with recent development and gain new knowledge. The implementation of such solution will enable developers specially newcomers to overcome package management-related issues, thereby improving overall software development experience.

# DATA AVAILABILITY

The datasets used in this study are comprehensively described in the 'Data Collection' section. Our dataset is made publicly available through this link: https://zenodo.org/record/8246509. It contains the following items: (i) a dataset of 14,330 Twitter ID interlinked with the GitHub ID of npm maintainers and 39,425 npm ecosystem-related tweets posted by them, and (ii) a sample dataset of 1,176 tweets and their identified topic of tweets, nature, and sentiment related to the npm ecosystem.

### Funding

This project is funded by the research cell of Noakhali Science and Technology University, Bangladesh under grant number: NSTU/RC-CS-04/T-23/46 and NSTU/RC-CS-01/T-23/43 and also by the Universitas Muhammadiyah Surakarta, Indonesia under grant number: 145.4/A.3-III/LRI/VII/2022. There was no additional external funding received for this study. The funders had no role in study design, data collection and analysis, decision to publish, or preparation of the manuscript.

### Grant Disclosures

The following grant information was disclosed by the authors:
Noakhali Science and Technology University, Bangladesh: NSTU/RC-CS-04/T-23/46 and NSTU/RC-CS-01/T-23/43.
Universitas Muhammadiyah Surakarta, Indonesia: 145.4/A.3-III/LRI/VII/2022.

### Competing Interests

The authors declare that they have no competing interests.

### Author Contributions

- Syful Islam conceived and designed the experiments, performed the experiments, analyzed the data, performed the computation work, prepared figures and/or tables, authored or reviewed drafts of the article, and approved the final draft.
- Yusuf Sulistyo Nugroho conceived and designed the experiments, performed the experiments, analyzed the data, performed the computation work, prepared figures and/or tables, authored or reviewed drafts of the article, and approved the final draft.
- Chy. Md. Shahrear performed the experiments, prepared figures and/or tables, authored or reviewed drafts of the article, and approved the final draft.
- Nuhash Wahed performed the experiments, prepared figures and/or tables, authored or reviewed drafts of the article, and approved the final draft.
- Dedi Gunawan performed the experiments, analyzed the data, prepared figures and/or tables, authored or reviewed drafts of the article, and approved the final draft.
- Endang Wahyu Pamungkas performed the experiments, analyzed the data, prepared figures and/or tables, authored or reviewed drafts of the article, and approved the final draft.

- Mohammed Humayun Kabir conceived and designed the experiments, analyzed the data, prepared figures and/or tables, authored or reviewed drafts of the article, and approved the final draft.
- Yogiek Indra Kurniawan performed the experiments, prepared figures and/or tables, authored or reviewed drafts of the article, and approved the final draft.
- Md. Kamal Uddin analyzed the data, prepared figures and/or tables, authored or reviewed drafts of the article, and approved the final draft.

## Data Availability

The dataset is available at Zenodo: Syful Islam, Yusuf Sulistyo Nugroho, Akib Shahrear, Nuhash Wahed, Dedi Gunawan, Endang Wahyu Pamungkas, Mohammed Humayun Kabir, Yogiek Indra Kurniawan, & Md. Kamal Uddin. (2023). An Empirical Study of Software Ecosystem Related Tweets by npm Maintainers. https://doi.org/10.5281/zenodo.8246509.

## Supplemental Information

Supplemental information for this article can be found online at http://dx.doi.org/10.7717/peerj-cs.1669#supplemental-information.

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
