# Peer review of "An empirical study of software ecosystem related tweets by npm maintainers"

_PeerJ Computer Science, doi:10.7717/peerj-cs.1669_

## Round 0.1 · original submission · Major Revisions

All three reviewers give high-quality comments. In general, most of the comments think that some parts of this paper are confusing, and the author should make revisions point by point according to the reviewers' comments, especially to improve the paper's clarity. For example, two reviewers expressed concerns about the description, "we prepare a sample dataset keeping 99% confidence". One of the reviewers believes that the author did not analyze the retweet and comment content corresponding to the tweet, and the author should consider this suggestion. In addition, I suggest the author use more diverse references to emphasize their research motivation and background, which will further illustrate the ability of social networks to serve various application scenarios. For example, social media as a social sensor has been widely used in natural disaster emergency management, intelligent transportation, etc. The author should make a major revision according to the comments.

Reviewer 1 ·

Basic reporting

The idea of the paper is clear and straightforward. The writing and presentation of the paper are good.

Experimental design

It is a cool study to look at Twitter as Question and Answer platform for npm. However, there are several major points to make the study with more useful implications. I have major concerns regarding specific approaches to the results. Next, I detail them.

- Motivation. The paper states that issues related to npm are discussed on Twitter. However, the motivation is weak. Why do we need Twitter specifically when other platforms are away richer and better structured (e.g., StackOverflow)? More on this is related to security issues related to npm packages. Some popular security events are sometimes discussed on Twitter more than on other platforms. Please check this and see if it can help to support the study. In fact, I missed anything related to npm security issues coming from tweets. Is that due to dataset limitation? As far as I know, npm used to be a prime target of attacks, see the following papers; they may be helpful for discussion and supporting the motivation of the study.
[1] Alfadel, Mahmoud, Nicholas Alexandre Nagy, Diego Costa, Rabe Abdalkareem, and Emad Shihab. "Empirical analysis of security-related code reviews in npm packages." Journal of Systems and Software (2023): 111752.
[2] Zimmermann, Markus, Cristian-Alexandru Staicu, Cam Tenny, and Michael Pradel. "Small World with High Risks: A Study of Security Threats in the npm Ecosystem." In USENIX security symposium, vol. 17. 2019.
[3] Alfadel, Mahmoud, Diego Elias Costa, Emad Shihab, and Bram Adams. "On the Discoverability of npm Vulnerabilities in Node. js Projects." ACM Transactions on Software Engineering and Methodology (2022).

- Identifying core developers of npm. The number of identified npm core developers seems to be based on the contributors to the package development. As far as I can tell, npm currently has more than 1.7 million packages, and hence the number of developers could be hundreds of thousands. Is this the entire ecosystem of npm developers? Were there any missing IDs on GitHub?

- RQ1 approach. What is the draft list of discussions and are they validated on the topics of twitter. The paper should discuss the motivation of using the list. Also, what is the exact score of the Kappa agreement?

- Line 184 in RQ2 approach: "the topics were revised ...into a larger category.." It is not clear how the abstraction was done. Please clarify it and support it with an example.

- RQ2 approach. The tweet is classified into Question, Answer, Information, and Other. Why the Answer class should be different from the Question class. It does not make send to me as both are related to each other.

- RQ3 approach. The approach of classifying the text into sentiments is not well-executed. First of all, the approach is fully manual, and this can be subjective. I am wondering why the paper does not use prior approaches that can automatically classify the text into different well-defined sentiments, e.g., TF-IDF vectorization. This will help even to classify a larger text. The manual analysis can support verifying some false positives after using the automatic approach.

Validity of the findings

Please see my comments on the Experimental design section above.

Additional comments

- The implications are generic and come without evidence. I would like the authors to expand on this. For example, the paper states (in Implication and Recommendation section) that developers should consider Twitter as an important tool for their development activities. Though this seems interesting, it comes without an evaluation. There is a need for input from npm developers. Can the author conduct a survey with a set of representative npm developers to run their findings and implication by them and support their implications?

Cite this review as

·

Basic reporting

The authors study the use of Twitter by npm core software developers to understand the value of Twitter to software packaging ecosystems.

Overall, this article is easy to read and well-structured, the authors are aware of the existing literature and have summarized their contributions and distinguished this paper with the previous ones.

I have one suggestion regarding the use of "npm core developers".
In the data collection section, "core developers" are identified as developers who "perform at least one pull request on software packages published on npm", this definition may be confused with previous papers discussing core developers (for example, A case study of open source software development: the Apache server by Mockus and others), where they define core developers as those who make a significant amount of contributions and this definition has been widely adopted in other studies.
It would reduce the confusion if the authors can term the study subjects as npm maintainers (or npm maintenance community)

Experimental design

While there has been many existing research studying the topics/contents of tweets about open-source, the authors specifically target the npm package maintainer community and understand their twitter usage. I think it is novel from previous research as it provide a detailed, fine-grained perspective on a smaller group and can provide more straightforward implication to the targeted audience.
Overall, the authors have summarized the previous works well and described their unique contributions.

Authors mostly use qualitatively coding in this study to capture the topic, nature and sentiment of tweets, the coding was conducted by more than one author and validated by the other two authors. A Kappa score was computed to evaluate the validity of coding.

I have two suggestions:
1. I suggest the authors can report the value of kappa score for each questions, as it is not listed in the paper right now.
2. I suggest the authors can be more clear about their sampling strategies. In Data Collection Section, step 3. The authors said "we prepare a sample dataset keeping 99% confidence with interval 3", it is hard to read and requires explanation (and reference).

Validity of the findings

The replication packages are provided with data available, and the validity are discussed in the validation section. I didn't see clear weakness in this part.

Reviewer 3 ·

Basic reporting

The study focuses on tweets from »core software developers«. In the introduction, clearly state what is meant by this term. Later in the study, in the Method section (lines 157-159: »/.. To identify core developers, we only select those who performed at least one pull request on software packages published on npm ecosystem .../«), this term is somewhat explained; however, I would suggest introducing the phrase earlier in the paper. In addition, is such an interpretation coined by the authors, or has it been used before in prior research?

In the introduction, what needs to be added is a short description of the methods used, preferably after the problem statement (lines 57-67) and before the results with contributions (lines 68-82). This will help the reader understand how the summarized results were gathered.

The Related Work section (lines 96-126) needs to be longer and introduce all relevant prior studies. The authors should improve this section by extending it. For instance, the main focus of this study is also on npm packages that can be found in the npm package registry. Many empirical studies focus on Javascript packages from npm from several perspectives, such as mining software repositories, software quality assurance, and security threats. This would put this work in the larger context of other works done on the npm ecosystem.

In Figure 2, the used icons are scaled proportionally, and the figure is missing a step (Step 1, i.e., identification of core developers), making the figure not in accordance with the description provided within the caption and the text.

The percentage values in Figure 6, Figure 7, and Figure 9 are hard to read.

In the Method section, I would suggest the authors to discuss how the analysis of tweets was performed regarding the topics, nature, and sentiments. Currently, the methods are presented alongside the results, making the paper harder to read and leading to some repetitions for methods employed over all three RQ.

I would also suggest the authors to consider using more uniform namings as several different ones are now used; for example, in the context of RQ1, the categories are described as topics of tweets/tweet categories; in the context of RQ2, the nature of tweets/tweet characteristics. More consistent namings make the paper easier to understand for the reader in order to know to which categorization the authors are referring, e.g., tweet topics and tweet formulation. This should also be reflected in the dataset, available as a part of the replication package.

The authors should recheck the replication package to make sure everything is included. For instance, in the file »Analysis_scripts.txt«, the content stops in the middle of a sentence (»/... For better visualization, we used latex packages such as«). As regards the scripts, it is understandable that API keys that the authors used should not be made public. Still, the rest of the scripts (with a placeholder for any API keys used) should be provided to speed up possible replications. In both the main and sample datasets, empty lines of Tweets are included; it might be good to eliminate them to improve the readability, especially of the XSLX sample dataset, so that each row corresponds to one data point in the dataset. In the current form, several rows are empty. In addition, I would advise to re-read the provided content in the replication packages, as there are multiple spelling and grammar mistakes, such as »for the shake of privacy«.

In general, the English language is adequate. Still, some minor improvements are needed, e.g., Twitter is capitalized in some cases and others not, and some typos (»shake« -> »sake«).

Experimental design

The last part of the research gap in the Introduction section (lines 53-57: »/... However, no study has been conducted that examines the tweets shared by developers to determine the challenges developers face, including the topic of discussion, nature, and sentiments associated with using software packaging ecosystems, as well as how they differ from question-and-answer sites. To bridge this gap ... /«) is not addressed in this paper. It should be made more transparent that the present study does not compare tweets with question-and-answer sites in any way.

In the results part of the Introduction section (line 68: »Our study demonstrated that Twitter contains useful information for the npm developer community.«), I would argue that the usefulness is never assessed in the study by any means, not by surveying developers or examining the engagement of the tweets. I do agree that the tweets might be useful. However, this is not something the study assessed; thus, it might be too strong of a statement.

The RQ1 might be misleading (lines 139-141: »/... What are the main topics of interest for npm core developers on Twitter? .../«). Instead, is not the study interested in what are the main topics that the core developers communicate on Twitter?

The presentation of results in the context of RQ2 is sound. However, the posed RQ might be misleading (lines 142-143: »/... What is the nature of the discussion that happens among npm core developers on various topics in the form of question and answers? ../«). What exactly do the authors mean by the last part,»/... in the form of question and answers. .../«. Based on the coding scheme used, it seems the tweets can be posted as information, questions, answers, or others. Should the RQ be formulated as: »What is the formulation nature of tweets communicated by npm core developers on various topics?«?

In addition to the covered topics by the three RQ, many other research directions could (and should) be made for the paper to aid in understanding the perceptions and opinions of core developers via the means of tweets communicated on Twitter. For instance, are the core developers referring to npm packages that they are actively contributing to; is there a difference in sentiment based on their evolvement; is there a link between package popularity/quality and tweets, the usefulness of different types of tweets, etc.?

As for the data collection part of the Method section (lines 169-171: »/.. we prepare a sample dataset keeping 99% confidence with 171 intervals 3. ../«), the authors should elaborate more on what that means. In addition, in the Results section, it is mentioned several times that the sample is statistically representative. How was that achieved?

The annotation process needs to be made clearer. What happens if the two final annotators do not agree when annotating tweets based on the topics or/and tweet formulation? Based on the provided data, in the end, each data point has one label for each categorization.

Why are not Tweets that are not in Englished removed from the dataset (N=42)? It seems such a tweet could not be categorized and considered in any of the RQ.

In the Method section of the RQ1 (lines 190-192: »/.. The Kappa score of our study is satisfactory after two rounds of 191 manual annotations by two authors. ../«), it should be clearly stated that was the score was as »satisfactory score« is too subjective. The same goes for RQ2 and RQ3.

Validity of the findings

I would like the authors to further discuss their results by providing additional more in-depth analyses, for instance, by extracting keywords or conducting topic modeling of each tweet topic/formulation.

One limitation the authors should have discussed is that the study is conducted only on tweet posts, not any other data related to tweets, such as comments or reposts.

Additional comments

Not all figures were made available to the reviewers in supplement materials.

Cite this review as

---

## Round 0.2 · Minor Revisions

The authors have addressed most of the concerns raised in the previous version. However, it is noted that the references utilized by the authors to substantiate the motivations behind their Twitter research appear somewhat dated (2011, 2014). To better address R1.1, it is recommended that the authors reinforce their motivation with more recent references to demonstrate that this social media-based topic is still valuable, as previously highlighted in my comments on the prior version. Furthermore, there are a few minor reviewer comments that also need attention.

Reviewer 1 ·

Basic reporting

I would like to express my gratitude to the reviewer for their diligent efforts in addressing the reviews. Nonetheless, I kindly request the consideration of the following minor observations:

R1.3: In the context of identifying core developers, it is essential to note that the requirement to link a Twitter account to a GitHub profile may result in the inadvertent exclusion of developers who do not have their Twitter accounts linked. Consequently, I recommend considering this potential limitation as a threat to validity.

R1.6: Given the authors' acknowledgment of the subjectivity in manual analysis, I would also propose incorporating this aspect into the discussion of potential threats to the paper's validity.

Experimental design

NA

Validity of the findings

NA

Additional comments

NA

Cite this review as

Reviewer 3 ·

Basic reporting

No comment.

Experimental design

No comment.

Validity of the findings

No comment.

Cite this review as

---

## Round 0.3 · accepted · Accept

The authors have improved the quality of their manuscript based on the reviewer's valuable comments, and all reviewers recommend accepting their paper.

Reviewer 1 ·

Basic reporting

I'd like to thank the authors for their great effort. I have no further comments on the paper.

Experimental design

NA

Validity of the findings

NA

Additional comments

NA

Cite this review as